# Beamforming with 1 × *N* Conformal Arrays

**DOI:** 10.3390/s22176616

**Published:** 2022-09-01

**Authors:** Irfan Ullah, Benjamin D. Braaten, Adnan Iftikhar, Symeon Nikolaou, Dimitris E. Anagnostou

**Affiliations:** 1Department of Electrical and Computer Engineering, COMSATS University Islamabad, Abbottabad Campus, Abbottabad 22060, Pakistan; 2Electrical and Computer Engineering Department, North Dakota State University, Fargo, ND 58102, USA; 3Department of Electrical and Computer Engineering, COMSATS University Islamabad, Islamabad 45550, Pakistan; 4Frederick Research Center, Nicosia 1303, Cyprus; 5Department of Electrical Computer Engineering and Informatics, Frederick University, Nicosia 1036, Cyprus; 6Institute of Signals, Sensors and Systems, Heriot Watt University, Edinburgh EH14 4AS, UK

**Keywords:** beam-forming antenna array, conformal surface, self-adapting

## Abstract

The rapid growth of wireless spectrum access through cellular and IoT devices, for example, requires antennas with more capabilities such as being conformal and self-adapting beamforming. In this paper, the adaptive beamforming patterns of microstrip patch antenna arrays on changing flexible (or conformal) curved surfaces are developed by deriving array coefficients based on the projection method that includes the mutual coupling between elements. A linear four-element microstrip patch antenna array is then embedded on two deformed conformal surfaces to investigate the projection method for desired beamforming patterns. The generated beamforming radiation patterns using the computed weighting coefficients are validated with theoretical equations evaluated in MATLAB, full-wave simulations in HFSS and measurement results. The measured results of the fabricated system agree with the simulated results. Furthermore, new guidelines are provided on the effects of mutual coupling and changing conformal surfaces for various beam-forming patterns. Such demonstrations pave the way to an efficient and robust conformal phased-array antenna with multiple beam forming and adaptive nulling capabilities.

## 1. Introduction

More sophisticated wireless technology will certainly be required to meet the demand of an increasing number of wireless users and IoT devices throughout the world accessing the spectrum. One major component of a wireless system that will fulfill this need of spectrum access is a smart antenna [1,2,3,4,5,6,7,8,9,10]. This is because a smart antenna can measure the surrounding electromagnetic environment and steer the main lobe of the pattern towards the desired users and move the nulls of the antenna toward unwanted noise sources; thus, making the antenna adaptive and increasing the capacity of the overall system in real time. Furthermore, to increase the coverage and capacity of multifunctional wireless platforms, conformal smart antennas are being placed in unique locations such as buildings, vehicles and aircraft [9,10]. In many of these locations, however, the conformal smart antennas are subjected to forces that may deform the topology of the antenna and thus degrade the performance.

In the area of conformal antennas, researchers have studied how the radiation pattern of a conformal antenna changes as it is deformed in various ways [11,12,13,14,15,16]. It has been shown that the overall gain of an antenna array can be reduced by as much as 25 dBi in a particular direction without appropriate phase and amplitude compensation [16]. To mitigate some of the unwanted effects of deformation on the overall gain of the array, several compensation techniques have been proposed [17,18,19,20,21,22,23,24,25,26]. The aforementioned literature study shows that the main focus is on the broadside radiation pattern correction of a conformal antenna array attached to a changing surface (i.e., vibrating surface) using composite amplitude and phase errors measurements, mechanical calibration methods, hardware sensor monitoring and the null points adjusting method. Then again, the previous work (with the exceptions of [27,28,29,30]) was developed under the assumption that the main lobe direction of the radiation pattern was fixed and the location of the nulls for beamforming was not defined.

Thus, the objective of this work is to study the theory of beamforming for conformal antenna arrays attached to the changing wedge- and cylindrical-shaped surfaces in Figure 1a,b for a specific main lobe direction and null locations for the first time. Moreover, the changing surface effects on the radiation properties of conformal beamforming antennas will be explored. In particular, this work will focus on the beamforming for the one dimensional 1 × *N* arrays on the wedge- and cylindrical-shaped surfaces in Figure 1a,b, respectively. For both Figure 1a,b, the location of each antenna element on the conformal surface is shown as a black dot and denoted as $A_{\pm n}$ where *N* is assumed to be even and n=−N/2,…,−1, 1,…,N/2. It should be mentioned that similar work could be carried out for an odd value of *N*. Moreover, the wavefront is denoted as a grey dotted line and the direction of propagation is shown with a black arrow. The wavefront is shown away from the origin for illustration purposes only.

## 2. Theory of Beamforming Antennas on Changing Conformal Surfaces

### 2.1. Beamforming of Antenna Array on a Singly Curved (Wedge Shaped) Surface

For this work, the 1 × 4 array (i.e., *N* = 4) on the singly curved (wedge)-shaped surface shown in Figure 2a with a bend angle of $\theta_{b}$ will be considered first. The spacing between adjacent antenna elements lying along the wedge is denoted as *d* and the location of element $A_{\pm n}$ is denoted as $\left(x_{\pm n},z_{\pm n}\right)$. The motivation for using a beamforming antenna is the feature of being able to generate a main beam in a desired direction while simultaneously defining nulls in the radiation pattern in other directions. The main lobe can then be used to communicate with a desired user in a particular direction and the nulls can be used to reduce the incoming signal from users not of interest. For this work, the angle of the signal of interest will be denoted as $\theta_{SOI}$ and one user of interest will be assumed while the angle of the n^th^ user not of interest will be denoted as $\theta_{SNOI_{n}}$. Because a four-element array is being considered here, three nulls at $\theta_{SNOI_{1}}$, $\theta_{SNOI_{2}}$ and $\theta_{SNOI_{3}}$ are defined. Now, the goal is to define appropriate array weights that can be used to drive the four-element array in Figure 1a,b to give a main lobe at $\theta_{SOI}$ and three nulls at $\theta_{SNOI_{1}}$, $\theta_{SNOI_{2}}$ and $\theta_{SNOI_{3}}$. Furthermore, to determine how the shape of the wedge changes the weights, they are to be determined for various bend angles $\theta_{b}$ of the wedge.

To compute the surface-dependent array weights, the projection method [31] will be used along with the matrix method for computing antenna weights defined in [1]. Since a maximum radiation in the direction of $\theta_{SOI}$ and nulls in the directions of $\theta_{SNOI_{1}}$, $\theta_{SNOI_{2}}$ and $\theta_{SNOI_{3}}$ are desired, the array will be considered as a transmitter. First, suppose the antenna array is radiating in the direction of $\theta_{SOI}$, as shown in Figure 2a. One method used to provide a field in the direction of $\theta_{SOI}$ is to ensure that the fields radiated from elements $A_{-2}$, $A_{-1}$, $A_{1}$ and $A_{2}$ reach at the reference wavefront with equal phases, thereby resulting in a broad-side pattern to the wavefront. To ensure that these fields arrive with the same phase, the voltage phase driving each individual element can be adjusted with a phase shifter. The amount of phase required is equal and opposite to the sign of the phase introduced by the free-space propagation from the antenna element to the wavefront. To compute this distance, the values for $\Delta_{-2}$, $\Delta_{-1}$, $\Delta_{1}$ and $\Delta_{2}$ in Figure 2a, which denote the distance from the elements $A_{-2}$, $A_{-1}$, $A_{1}$ and $A_{2}$, respectively, to the wavefront (or the projected elements on the wavefront), need to be computed. Furthermore, the expressions for $A_{-2}$, $A_{-1}$, $A_{1}$ and $A_{2}$ should be written in a general manner including surface dependent position of each antenna element and the bend angle $\theta_{b}$. A similar argument can be made for the case when the array is transmitting a null in the direction of $\theta_{SNOI_{n}}$, which is shown in Figure 2b. For this case, the values of $\Delta_{\pm n}$ still need to be computed.

### 2.2. Computing the Distance to the Projected Elements on the Wave Front

For this work, the values of $\theta_{SOI}$ and $\theta_{SNOI_{n}}$ are between −π/2 and π/2. Because of this, the problem will be broken down into two cases. For the first case, $-\theta_{b}\;\leq \;\theta_{SOI\left(SNOI_{n}\right)}\;\leq \;\theta_{b}$. For these angles of $\theta_{SOI\left(SNOI_{n}\right)}$, the projected elements on the wavefront are all outside of the wedge-shaped surface. For the second case, $\theta_{b}\;\leq \;\theta_{SOI\left(SNOI_{n}\right)}\;\leq \;\pi /2$ or $-\pi /2\;\leq \;\theta_{SOI\left(SNOI_{n}\right)}\;\leq -\theta_{b}$. For these angles of $\theta_{SOI\left(SNOI_{n}\right)}$, half of the projected elements are outside of the wedge-shaped surface and half are inside of the surface. This is because when $\theta_{SOI\left(SNOI_{n}\right)}\;\geq \;\theta_{b}$ or $\theta_{SOI\left(SNOI_{n}\right)}\;\leq -\theta_{b}$, the projected elements for $A_{1}$ and $A_{2}$ or $A_{-1}$ and $A_{-2}$, respectively, are inside of the wedge-shaped surface in Figure 2a,b.

#### 2.2.1. Case 1: Computing $\Delta_{\pm n}$ for $-\theta_{b}\leq \theta_{SOI\left(SNOI_{n}\right)}\leq \theta_{b}$

Using the notation in Figure 2a,b and several identities, the projected distance from antenna elements on the wedge to the reference wavefront is given by:(1)$$\Delta_{\pm n}=d\frac{\left|n\right|}{2n}\left(2\left|n\right|-1\right)\sin\;\left(\theta_{b}\;\mp \;\theta_{SOI\left(SNOI_{n}\right)}\right)$$
for $0\;\leq \;\theta_{SOI\left(SNOI_{n}\right)}\;\leq \;\theta_{b}$ and
(2)$$\Delta_{\pm n}=d\frac{\left|n\right|}{2n}\left(2\left|n\right|-1\right)\sin\;\left(\theta_{b}\;\pm \;\left|\theta_{SOI\left(SNOI_{n}\right)}\right|\right)$$
for $-\theta_{b}\;\leq \;\theta_{SOI\left(SNOI_{n}\right)}\;\leq \;0$. Notice that the expressions in (1) and (2) are written as a function of the flexing (bend) angle $\theta_{b}$ of the wedge and the location of each antenna element, indicating the generality of the expressions.

#### 2.2.2. Case 2: Computing $\Delta_{\pm n}$ for $\theta_{b}\leq \theta_{SOI\left(SNOI_{n}\right)}\leq \pi /2$ or $-\pi /2\leq \theta_{\mathrm{SOI}\left({\mathrm{SNOI}}_{n}\right)}\leq -\theta_{b}$

Next, for the remaining values of $\theta_{SOI\left(SNOI_{n}\right)}$, the projected distance from each antenna element in the array to the reference wavefront can be calculated as:(3)$$\Delta_{\pm n\;}=d\frac{\left|n\right|}{2n}\left(2\left|n\right|-1\right)\sin\;\left(\theta_{SOI\left(SNOI_{n}\right)}\;\mp \;\theta_{b}\right)$$
for $\theta_{b}\;\leq \;\theta_{SOI\left(SNOI_{n}\right)}\;\leq \;\pi /2$ and
(4)$$\Delta_{\pm n}=d\frac{\left|n\right|}{2n}\left(2\left|n\right|-1\right)\sin\;\left(\left|\theta_{SOI\left(SNOI_{n}\right)}\right|\;\pm \;\theta_{b}\right)$$
for $-\pi /2\;\leq \;\theta_{SOI\left(SNOI_{n}\right)}\;\leq \;\theta_{b}$. Notice that (3) and (4) are also written in terms of $\theta_{b}$ of the wedge and the position of each antenna element in a general fashion.

### 2.3. Computing the Radiation in the Direction of $\theta_{SOI}$, $\theta_{SNOI_{1}}$, $\theta_{SNOI_{2}}$ and $\theta_{SNOI_{3}}$

Next, the field from the array in the direction of $\theta_{SOI}$ (signal of interest), and $\theta_{SNOI_{1}}$, $\theta_{SNOI_{2}}$ and $\theta_{SNOI_{3}}$(signals of no interest) on the singly curved surface (wedge) is computed with the Array Factor $\left(AF_{w}\right)$ expression in [15]:(5)$$AF_{w}\left(\theta ,\phi \right)=\overset{-1}{\underset{n=-2}{\sum }}F_{n,L}e^{jk\left[ux_{n}+vy_{n}+z_{n}\cos\theta \right]}+\overset{2}{\underset{n=1}{\sum }}F_{n,R}e^{jk\left[ux_{n}+vy_{n}+z_{n}\cos\theta \right]}$$
where
(6)$$F_{n,L}=w_{n}\cos\theta_{PL}e^{\pm jk\Delta_{\pm n}}$$
and
(7)$$F_{n,R}=w_{n}\cos\theta_{PR}e^{\pm jk\Delta_{\pm n}}$$

A spherical coordinates system is assumed in (5) with u=sinθcosϕ, v=sinθsinϕ, $\Delta_{\pm n}$ is defined in (1)–(4), $\left(x_{n},y_{n},z_{n}\right)$ is the location of the *n*^th^ array element and $w_{n}$ are the array excitation weights (amplitudes and phases) to drive the individual antenna elements. Furthermore, the element patterns for $A_{1}$ and $A_{2}$ are denoted as $e_{R}\left(\theta \right)=\;\cos\theta_{PR}$ and the element patterns for $A_{-1}$ and $A_{-2}$ are denoted as $e_{L}\left(\theta \right)=\cos\theta_{PL}$ where $\theta_{PR}$ and $\theta_{PL}$ are defined in Figure 2.

### 2.4. Computing the Array Weights for N = 4 Elements

The complex weighting functions (i.e., array weights) in (5) are of interest in this work because these are the array weights that need to be computed for beamforming of the array. To compute these weights, the matrix method for smart antennas presented in [1] will be used. Because the array studied here has four elements, four array weights are needed. This then requires a set of four equations with four unknowns that can be written as a square matrix. Then, the array weights can be computed using a matrix solver. The sum in (5) has four terms for *N =* 4 and to obtain four equations with four unknowns, (5) will be evaluated at each value of $\theta_{SOI}$, $\theta_{SNOI_{1}}$, $\theta_{SNOI_{2}}$ and $\theta_{SNOI_{3}}$. This then gives the following array factor matrix for antenna array on wedge-shaped conformal surface $AF_{w}$:(8)$$AF_{w}=\left[\begin{array}{c} \begin{array}{c} AF_{w}\left(\theta_{SOI}\right)\\ AF_{w}\left(\theta_{SNOI_{1}}\right) \end{array}\\ AF_{w}\left(\theta_{SNOI_{2}}\right)\\ AF_{w}\left(\theta_{SNOI_{3}}\right) \end{array}\right]$$
where $\theta =\theta_{SOI}$, $\theta_{SNOI_{1}}$, $\theta_{SNOI_{2}}$ or $\theta_{SNOI_{3}}$ and ϕ=0. Then, factoring out the array weights gives,
(9)$$AF_{w}=AW\;$$
where A is the array factor in (5) for antenna array on wedge-shaped conformal surface with factoring out the complex weights and
(10)$$W=\left[\begin{array}{c} \begin{array}{c} w_{-2}\\ w_{-1} \end{array}\\ w_{1}\\ w_{2} \end{array}\right]\;$$

Next, to ensure that the conformal antenna will have a main beam at the scan angle $\theta_{SOI}$, $AF_{w}\left(\theta_{SOI}\right)$ must equal 1 in (8), or $AF_{w}\left(\theta_{SOI}\right)=1$. Then, in order to provide nulls in the direction of $\theta_{SNOI_{1}}$, $\theta_{SNOI_{2}}$ and $\theta_{SNOI_{3}}$, $AF_{w}\left(\theta_{SNOI_{1}}\right)=AF_{w}\left(\theta_{SNOI_{2}}\right)=AF_{w}\left(\theta_{SNOI_{3}}\right)=0$ in (8). This can be written in matrix form in the following manner:(11)AW=C
where
(12)$$C=\left[\begin{array}{c} \begin{array}{c} 1\\ 0 \end{array}\\ 0\\ 0 \end{array}\right]$$
and AW is defined in (9). Finally, solving for the weights in (11) gives
(13)$$W=A^{-1}C$$

By setting the first element in (12) to be 1 and the rest of the elements to be 0, the array factor is forced to give nulls (or a zero field) analytically at the angles of $\theta_{SNOI_{1}}$, $\theta_{SNOI_{2}}$ and $\theta_{SNOI_{3}}$. Thus, the solution of (13) are the array weights required to give these pattern null features. Furthermore, since the values of $\Delta_{\pm n}$ in the array factor expression in (8) are written in terms of the wedge angle $\theta_{b}$ and the location of each antenna element, the weights computed with (13) are determined in a setting where the antenna, and hence the wedge, can change shape.

### 2.5. Computing the Array Weights for N Elements

The previous expressions were determined for *N* = 4 and can be generalized for more elements. More, specifically, (8)–(13) can be generalized in the following manner:(14)$$AF_{w}=\left[\begin{array}{c} AF_{w}\left(\theta_{SOI}\right)\\ .\\ .\\ .\\ AF_{w}\left(\theta_{SNOI_{1}}\right)\\ .\\ .\\ .\\ AF_{w}\left(\theta_{SNOI_{n}}\right) \end{array}\right]$$
where
(15)$$AF_{wn}\left(\theta ,\phi \right)=\overset{-1}{\underset{n=-\frac{N}{2}}{\sum }}F_{n,L}e^{jk\left[ux_{n}+vy_{n}+z_{n}\cos\theta \right]}+\overset{\frac{N}{2}}{\underset{n=1}{\sum }}F_{n,R}e^{jk\left[ux_{n}+vy_{n}+z_{n}\cos\theta \right]}$$
and $F_{n,L}$ and $F_{n,R}$ are defined in (6) and (7), respectively. Then, factoring out the array weights gives
(16)$$AF_{wn}=A_{n}W_{n}$$
where $A_{n}$ is the array factor in (15) for antenna array on wedge-shaped conformal surface with factoring out the complex weights and
(17)Wn=[w−N2…w−1 w1…wN2]T

Finally, the coefficients in (12) can be generalized as
(18)$$C_{n}={\left[c_{1}\;c_{2}\ldots c_{n}\right]}^{T}$$
and the weights can be computed using
(19)$$W_{n}=A_{n}{}^{-1}C_{n}$$

Equation (19) can then be used to solve for the array weights that will result in a main radiation pattern in the direction of $\theta_{SOI}$ and nulls in the directions of $\theta_{SNOI_{1}}$*,* $\theta_{SNOI_{2}}$, … $\theta_{SNOI_{\left(N-1\right)}}$ for an N-element array, on a changing wedge-shaped surface.

### 2.6. Beamforming of a 1 × 4 Array on a Cylindrical-Shaped Surface

Next, the complex weights for antenna array on the cylindrical curvature shown in Figure 2c are computed. Again, for illustration, the derivations will be carried out for N=4 elements. As with the wedge-shaped conformal surface, the projected distance from the antenna elements on the cylindrical surface to the transmitted wavefront at angle $\theta_{SOI\left(SNOI_{n}\right)}$ must be computed. Again, these distances are denoted as $\Delta_{\pm n}$ in Figure 2c. First, the distance from the point where the cylinder intersects the *z*-axis (denoted as point P in Figure 2c) to each antenna element is computed using the following equation
(20)$$h_{\pm n}=\sqrt{{\left(0-x_{\pm n}\right)}^{2}+{\left(r-z_{\pm n}\right)}^{2}},$$
where again $\left(x_{\pm n},z_{\pm n}\right)$ is the location of the *n*^th^ element on the cylindrical surface, $x_{\pm n}=r\cos\phi_{\pm n}$, $z_{\pm n}=r\sin\phi_{\pm n}$ and $\phi_{\pm n}$ is defined in Figure 2c. As with the wedge-shaped surface, the problem will be considered as two different cases.

The first case is for $\theta_{SOI\left(SNOI_{n}\right)}\;\geq \;0$ and the second case is for $\theta_{SOI\left(SNOI_{n}\right)}\;\leq \;0$. The first case is shown in Figure 2c. The projected elements of $A_{-2}$, $A_{-1}$ and $A_{2}$ in the direction of $\theta_{SOI\left(SNOI_{n}\right)}$ are outside of the cylindrical surface and the projected element of $A_{1}$ in the direction of $\theta_{SOI\left(SNOI_{n}\right)}$ is inside of the cylindrical surface. Because some projected elements are outside of the cylinder and some are inside, each case will be broken into two parts. The first part will compute the distance from the elements $A_{-1}$ and $A_{\pm 2}$ to the projected elements on the wavefront and the second part will compute the distance from the element $A_{1}$ to the projected element on the wavefront (as shown in Figure 2c).

#### 2.6.1. Case 1: $\theta_{SOI\left(SNOI_{n}\right)}\geq 0$

Next, using (20) and several trigonometric identities, the projected distance from the elements $A_{-1}$ and $A_{\pm 2}$ to the transmitted wavefront is calculated as:(21)$$\Delta_{\pm n}=h_{\pm n}\sin(\mp \theta_{SOI\left(SNOI_{n}\right)}+\theta_{\pm n})$$
where $\theta_{\pm n}=\cos^{-1}\left|x_{\pm n}/h_{\pm n}\right|$. Then, for the element $A_{1}$ the distance to the projected element can be computed as:(22)$$\Delta_{+n}=h_{+n}\sin\;\left(\theta_{SOI\left(SNOI_{n}\right)}-\theta_{+n}\right)$$

#### 2.6.2. Case 2: $\theta_{SOI\left(SNOI_{n}\right)}\leq 0$

Again, using (20) and several trigonometric identities, the distance from the elements $A_{1}$ and $A_{\pm 2}$ to the projected elements can be computed as:(23)$$\Delta_{\pm n}=h_{\pm n}\sin\;(\pm \left|\theta_{SOI\left(SNOI_{n}\right)}\right|+\theta_{\pm n})$$
where $\theta_{\pm n}=\cos^{-1}\left|x_{\pm n}/h_{\pm n}\right|$. Then, for the element $A_{-1},$ the distance to the projected element can be computed as:(24)$$\Delta_{-n}=h_{-n}\sin\;\left(|\theta_{SOI\left(SNOI_{n}\right)}|\;-\theta_{-n}\right)$$

Several comments can be made about (21)–(24). As with the wedge-shaped surface, these expressions have been written in a general manner in terms of antenna position on the cylindrical surface and the radius of curvature. Moreover, special care should be taken when implementing these equations if $\left|\theta_{SOI\left(SNOI_{n}\right)}\right|\;\leq \;\theta_{\pm n}$. This is because for these projected elements, angles on the wavefront will be outside of the cylindrical surface. In this case, the distance to the wavefront can be computed using (21) or (23) (depending on whether $\theta_{SOI\left(SNOI_{n}\right)}\;\leq \;0$ or $\theta_{SOI\left(SNOI_{n}\right)}\;\geq \;0$ and the computations of (22) and (24) are not required.

### 2.7. Computing the Array Weights for N = 4 Elements

Next, as with the antenna array on the wedge-shaped conformal surface, the array weights can be computed using (13). For the computations of these array weights on the cylinder, the distance from the elements to the wavefront should be computed using (21)–(24).

### 2.8. Computing the Array Weights for N Elements

Finally, (19) can be used (with the updated distances computed using (21)–(24)) to compute the array weights for an *N*-dimensional array on a cylindrical surface with radius *r*. As with the wedge-shaped surface, the expressions to compute the array weights have been written in a general manner that includes element spacing and the radius of the cylinder. This makes the technique presented here useful for an array attached to a changing cylindrical surface.

### 2.9. Computing the Weighting Coefficients with Mutual Coupling

For the previous derivations of (13) and (19), it was assumed that there was no mutual coupling between the elements. To model the mutual coupling, the work presented in [32] was considered. More specifically, the work in [32] proposed a model for the mutual coupling in adaptive arrays and demonstrated that the mutual coupling between the elements of an adaptive array can cause a significant degradation in the signal-to-interference-plus-noise ratio (SINR). The methods in [32] will be adopted here to model the mutual coupling effects on the radiation pattern of the conformal beamforming array. The coupling between the antenna elements can be modelled as an *N* + 1 port network, as shown in Figure 3. The antenna elements in the array are all terminated with $Z_{L}$ and denoted as ports 1, 2, … *N*. The antenna port driven by a voltage source $V_{s}$ is denoted as *N* + 1. The port driven with $V_{s}$ is the representation of the transmitted (or incoming) signals at angles $\theta_{SOI}$ or $\theta_{SNOI_{n}}$. Then, using the Kirchhoff relations for the *N* + 1 terminal network, the voltage at the terminated port *n* can be written as:(25)$$V_{Tn}=I_{1}Z_{n,1}+I_{2}Z_{n,2}+\ldots +I_{n}Z_{n,m}+\ldots +I_{s}Z_{n,s}$$
where $Z_{n,m}$ is the mutual impedance between the *n*^th^ and *m*^th^ port, $I_{n}$ is the current going through the terminating load on the *n*^th^ port, $Z_{n,n}$ is the self-impedance of the *n*^th^ port and $Z_{n,s}$ represents the mutual coupling term between the driven element with $V_{s}$ and the *n*^th^ antenna element. Furthermore, the current at the *n*^th^ port can be written in terms of the terminal voltage and load impedance in the following manner:(26)$$I_{n}=\;\frac{-V_{Tn}}{Z_{L}}$$

Then, making use of the open circuit condition and removing the terminating impedances results in $I_{1}=I_{2}=\ldots I_{N}=0$. This simplifies (25) to $V_{Tn}=I_{s}Z_{n,s}$. Under these conditions, (25) represents the open circuit voltages at the *n*^th^ port caused by the mutual coupling between the driven element and the *n*^th^ port, and can be computed as $V_{Tn}=I_{s}Z_{n,s}V_{OCn}$. Next, substituting (26) into (25), making use of the open-circuit condition and writing (25) in matrix form results in the following:(27)$$Z_{c}V_{T}=V_{OC}$$
where
(28)$$Z_{c}=\left[\begin{array}{c} \begin{array}{cc} 1+\frac{Z_{1,1}}{Z_{L}} & \begin{array}{cc} \;\;\;\;\;\;\frac{Z_{1,2}}{Z_{L}} & \begin{array}{cc} \;\;\;\cdots & \frac{Z_{1,N}}{Z_{L}} \end{array} \end{array}\\ \frac{Z_{2,1}}{Z_{L}} & \begin{array}{cc} 1+\frac{Z_{2,2}}{Z_{L}} & \begin{array}{cc} \;\cdots & \frac{Z_{2,N}}{Z_{L}} \end{array} \end{array} \end{array}\\ \;\;\begin{array}{ccc} \begin{array}{cc} \vdots & \;\;\;\;\;\;\;\;\;\;\;\;\;\vdots \end{array}\;\;\;\;\;\;\; & \ddots & \;\;\;\;\vdots \end{array}\\ \begin{array}{lll} & \;\frac{Z_{N,1}}{Z_{L}} & \;\;\;\;\;\;\;\;\;\;\frac{Z_{N,2}}{Z_{L}}\;\;\begin{array}{cc} \;\ldots & 1+\frac{Z_{N,N}}{Z_{L}} \end{array} \end{array} \end{array}\right]$$
(29)$$V_{T}={\left[V_{T1\;}V_{T2}\ldots V_{\mathrm{Tn}}\right]}^{T}$$
and
(30)$$V_{OC}={\left[V_{\mathrm{OC}1}{\;V}_{\mathrm{OC}2\;\cdots }V_{\mathrm{OCn}}\right]}^{T}$$

The normalized impedance matrix $Z_{c}$ includes self- and mutual-terms, and can be determined from a 3D full wave simulator such as HFSS [33]. The open circuit voltage column matrix $V_{OC}$ represents the array weights (i.e., the complex voltages used to drive the beamforming conformal antenna) without including mutual coupling. This then results in $V_{OC}=W_{n}$ where $W_{n}$ is given in (19). The terminal voltage column matrix $V_{T}$ represents the array weights that include the mutual coupling and can be computed from (27) as follows:(31)$$V_{T}=Z_{c}{}^{-1}V_{OC}$$

Next, to write the array weights with mutual coupling in terms of $Z_{c}$, substitute $V_{OC}=W_{n}$ into (19) and rearranging gives the following expression:(32)$$A_{n}V_{OC}=C_{n}$$

Then, solving for $V_{OC}$ in (32) and substituting into (31) gives:(33)$$V_{T}=Z_{c}{}^{-1}A_{n}{}^{-1}C_{n}=W_{n}^{c}$$
where $W_{n}^{c}$ represents the new array weights with the coupling included in the computations. In the next Section, validation of $W_{n}$ using (19) and $W_{n}^{c}$ using (33) for various values of $\theta_{SOI}$ and $\theta_{SNOI_{n}}$ will be presented followed by the characteristics of the array weights for various conformal surfaces.

## 3. Validation with Analytical, Simulations and Measurement Results

In this Section, a beamforming array is used to validate the previously derived array weight expressions using simulations and measurements. More specifically the array weights are computed using (19) and (33) for the 1 × 4 element array on the singly curved wedge and cylindrical curvature surfaces, shown in Figure 2. Two different beam-formation patterns were considered, and each pattern was evaluated on three different conformal surfaces. The characteristics of each beam-formation pattern, denoted as pattern 1 and pattern 2, are mentioned in Table 1. Furthermore, the three conformal surfaces considered were the singly curved wedge surface with *θ_b_* = 30° and *θ_b_* = 45° and a cylindrical surface with *r* = 10 cm.

### 3.1. Four-Element Beamforming Array

For measurement purposes, the four-element beamforming array shown in Figure 4a was manufactured. The array consisted of connectorized voltage-controlled phase shifters, voltage-controlled attenuators, a four-way power divider, an amplifier and four microstrip patches designed to operate at 2.47 GHz. A picture of the attenuators, phase shifters and power divider of the manufactured array is shown in Figure 4b, and a picture of the four microstrip patches is shown in Figure 4c. Four individual microstrip patches were used for the convenience of placing the array on the various conformal surfaces. The phase shifters were manufactured by Hittite Microwave Corporation [34] (PN: HMC928LP5E) and the power divider, attenuators and amplifiers were manufactured by Mini-Circuits [35] (PNs: ZN8PD1-53-S+, ZX73-2500-S+ and ZX60-33LN-S+, respectively). Identical SMA cables were used to connect each patch to a port on the power divider.

### 3.2. Beamforming Results on the Conformal Wedge Antenna Array with θ_b_ = 30°

First measurements taken were for the 1 × 4 element beamforming array on the singly curved flexing wedge in Figure 2a for *θ_b_* = 30°, and the array weights were computed using (33). The inter-element spacing was 0.5 *λ* and a picture of the array being measured on the surface with a 2-port network analyzer in a fully calibrated anechoic chamber is shown in Figure 5a. The results from these measurements for both patterns summarized in Table 1 are shown in Figure 6a,b. Next, the four-element beamforming array was modelled in HFSS and the weights computed using (33) were used to drive the array. More specifically, the separate substrates, conducting layers and SMA connectors were modelling in HFSS to provide an accurate representation of the measurement setup in Figure 5. The radiation pattern from these simulations can also be seen in Figure 6a,b at 2.47 GHz. Then, for a third comparison both patterns were computed analytically using the array factor expression in (5) with the weights determined using (19). These results are also shown in Figure 6a,b. Finally, new weights that include the mutual coupling were computed using (33) and used in (5) to compute the radiation pattern analytically. These results are shown in Figure 6a,b. Overall, agreement between measurements, simulations and analytical computations (with both sets of array weights) is shown. The array weights for the results in Figure 6a,b are shown in Table A1 and Table A2 in Appendix A.

### 3.3. Beamforming Results on the Conformal Wedge Antenna Array with θ_b_ = 45°

Next, measurements were taken for the four-element beam-forming array on the singly curved flexing wedge in Figure 2a for *θ_b_* = 45°. The adjacent element spacing was again 0.5 *λ* and (33) was used to compute the new array weights. The results from these measurements for both patterns summarized in Table 1 are shown in Figure 7a,b. Next, the four-element beamforming array was simulated in HFSS with the array weights computed using (33) for *θ_b_* = 45° and the radiation pattern can also be seen in Figure 7a,b at 2.47 GHz. Then, for a third comparison, the expression in (19) was used to compute the analytical results shown in Figure 7a,b. Next, the four-element beamforming array was simulated in HFSS with the array weights computed using (33) for *θ_b_* = 45° and the radiation pattern can also be seen in Figure 7a,b at 2.47 GHz. Then, for a third comparison the expression in (19) was used to compute the analytical results shown in Figure 7a,b. New weights that include the mutual coupling were also computed using (33) for the new value of *θ_b_* = 45°. These weights were then used in (5) to compute the radiation pattern and these results are shown in Figure 7a,b. As with the *θ_b_* = 30° results, agreement between measurements, simulations and analytical computations is shown. The array weights for the results in Figure 7a,b are shown in Table A3 and Table A4 in Appendix A.

### 3.4. Beamforming Results on the Conformal Cylindrical Antenna Array with r = 10 cm

Finally, the same comparison between measurements, simulations and analytical computations was conducted for the four-element array on the cylindrical-shaped surface with *r* = 10 cm. The array being measured in the full anechoic chamber is shown in Figure 5b and the results are shown in Figure 8. It should be mentioned that in order to measure the array on a cylindrical surface, a sphere was used and the antenna elements were placed along the equator. This then resulted in an antenna shape similar to a cylindrical surface. Simulation results and analytical computations with and without the coupling are also shown in Figure 8 and overall agreement is shown. The array weights for the results in Figure 8 are shown in Table A5 and Table A6 in Appendix A.

## 4. Discussions

For the results in Figure 6 and Figure 7, the most agreement is between the measured results, the HFSS simulations and the analytical computations with the array weights including the mutual coupling. This illustrates the improved accuracy of the weights computed using (33).

Moreover, when comparing the pattern 2 results in Figure 6b and b, it is shown that more agreement between the results around −60° exist for the *θ_b_* = 45° surface than for the *θ_b_* = 30° surface. This is thought to be due to the more severe surface deformation that exists for the *θ_b_* = 45°. Overall though, the array weights computed using (19) and (33) have been shown to be accurate and the effects due to the mutual coupling and surface deformations on the radiation pattern have been demonstrated. Thus again, the weights computed using (33) can be used to model the coupling, and with proper optimization, an improved beam-formation could be achieved in a general setting that includes the mutual coupling between elements on a changing conformal surface.

The approach to beamforming presented in this paper has many applications ranging from IoT devices, 5G and mobile satellite communications. As devices in these spaces become smaller, phased array antennas are required to operate in the vicinity of scattering surfaces and other array elements, hence requiring knowledge of mutual coupling.

## 5. Dependence of Complex Weights on the Geometry of the Conformal Surface

Next, the array weights for a three-element array were computed analytically using (19). This was undertaken to show how the array weights are dependent on bend angle *θ_b_* and the radius of curvature *r*. It should be mentioned that = 3 was chosen because the problem becomes much more complex for *N* > 3 and requires a numerical analysis of the matrices in (19). After several algebraic steps it can be shown that the array weights can be written as:(34)$$w_{+1}=\frac{1}{F_{1,R}{}^{\Delta }-DF_{-1,L}{}^{\Delta }}$$
(35)$$w_{0}=1-w_{+1}\left[F_{1,R}\left(\theta_{SOI}\right)-DF_{-1,L}\left(\theta_{SOI}\right)\right]$$
and
(36)$$w_{-1}=-w_{+1}D$$
where
(37)$$D=\frac{F_{1,R}\left(\theta_{SNOI_{2}}\right)-F_{1,R}\left(\theta_{SNOI_{1}}\right)}{F_{-1,L}\left(\theta_{SNOI_{2}}\right)-F_{-1,L}\left(\theta_{SNOI_{1}}\right)}$$

$F_{1,R}{}^{\Delta }=F_{1,R}\left(\theta_{SOI}\right)-F_{1,R}\left(\theta_{SNOI_{1}}\right)$, $F_{-1,L}{}^{\Delta }=F_{-1,L}\left(\theta_{SOI}\right)-F_{-1,L}\left(\theta_{SNOI_{1}}\right)$ and $F_{n,L}$ and $F_{n,R}$ are defined in (6) and (7), respectively, with the weights factored out. For a small value of N the results in (34)–(36) show that the expression for the array weights can be quite complicated. Then again, it does also show the dependency of each weight on the angle $\theta_{SOI\left(SNOI_{n}\right)}$ and the location of each antenna element in the array.

To illustrate the behavior of the weights using (34)–(36) for various surfaces, the three-element array was considered on a wedge-shaped surface for various values of *θ_b_*. For these computations, the values of $\theta_{SOI}$ and $\theta_{SNOI_{n}}$ were $\theta_{SOI}=40^{{}^{\circ}}$, $\theta_{SNOI_{1}}=-45^{{}^{\circ}}$ and $\theta_{SNOI_{2}}=10^{{}^{\circ}}$. The amplitude and phase of the array weights computed using (34)–(36) are shown in Figure 9. The results show that for a fixed beam-formation there is a strong relationship between the array weights and the conformal surface. A similar derivation could be conducted to compute (33) for the array weights that include coupling, and the same relationship between the array weights and the surface deformation was shown to exist for the cylindrical surface.

## 6. Conclusions

Beamforming of a one-dimensional array on a changing singly curved wedge and cylindrical curvature surface was investigated. New matrices for computing the array weights that both do not include and include the mutual coupling between elements are presented. The computations of these array weights were then validated with the simulation and measurements of a four-element beamforming array. Overall agreement between the results was shown and the characteristics of a beamforming array on a conformal surface were discussed.

## Figures and Tables

**Figure 1 sensors-22-06616-f001:**
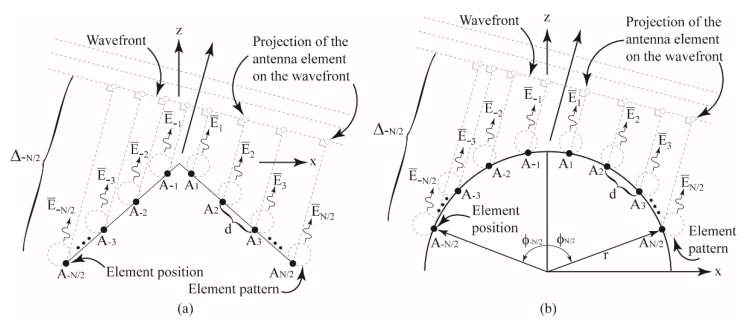
(**a**) Illustration of the antenna elements on a singly curved (wedge) surface and (**b**) illustration of the antenna elements on a cylindrical-shaped surface.

**Figure 2 sensors-22-06616-f002:**
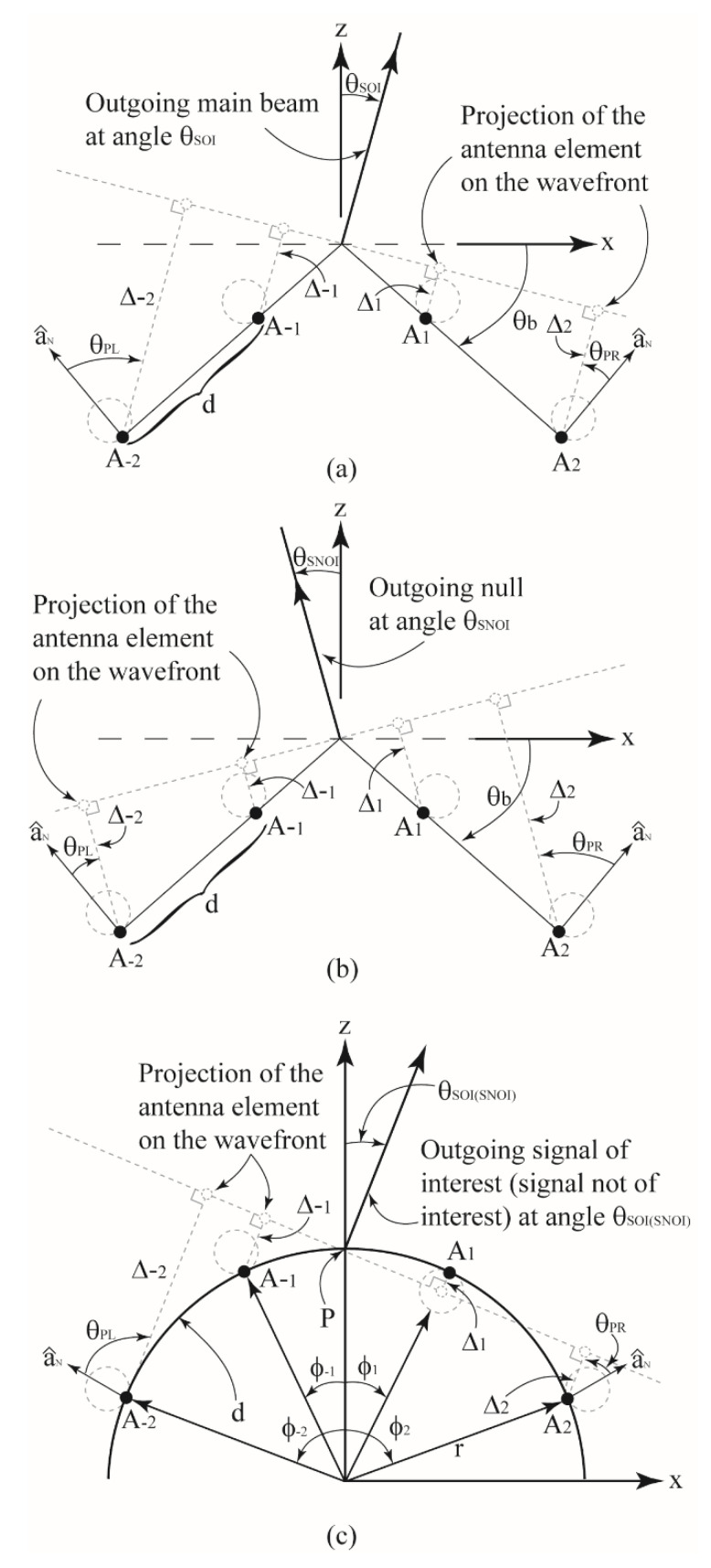
Geometrical illustration of the proposed projection method for (**a**) antenna array on a wedged-shaped surface with desired signal at angle *θ_SOI_*; (**b**) antenna array on a wedged-shaped surface with undesired signal at angle *θ_SNOI_* and (**c**) antenna array on a cylindrical curvature with desired (undesired) signal at angle *θ_SOI(SNOI)_*.

**Figure 3 sensors-22-06616-f003:**
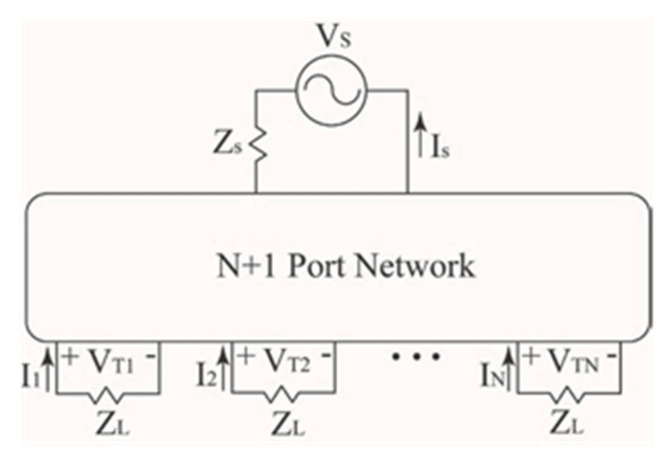
*N*-port network illustration of conformal array with a signal of interest at angle $\theta_{SOI}$ or $\theta_{SNOI_{n}}$.

**Figure 4 sensors-22-06616-f004:**
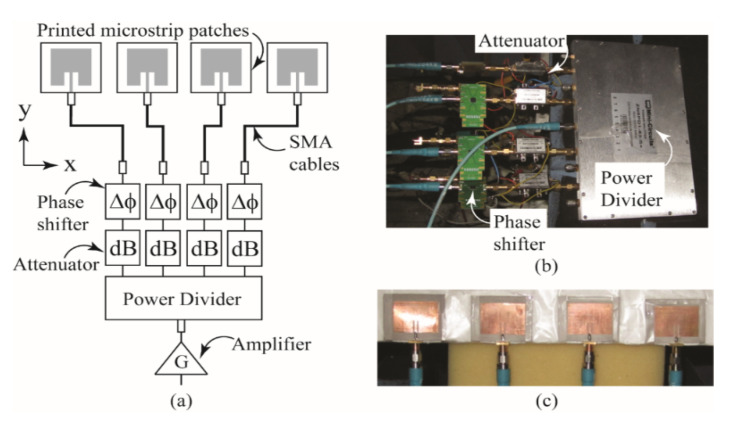
(**a**) Topology of the four-element beamforming array, (**b**) a picture of the power divider, voltage-controlled phase shifters and voltage-controlled attenuators used for measurements and (**c**) a picture of the microstrip patch elements used for attachment to conformal surfaces.

**Figure 5 sensors-22-06616-f005:**
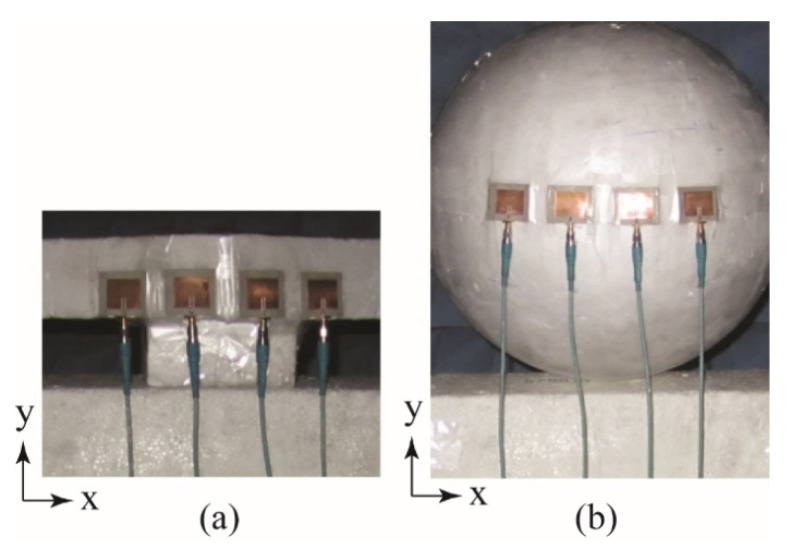
(**a**) Photograph of the four-element beamforming array being measured on the singly curved non-conducting wedge surface with *θ_b_* = 30° and (**b**) photograph of the four-element beamforming array being measured on the cylindrical-shaped surface with *r* = 10 cm.

**Figure 6 sensors-22-06616-f006:**
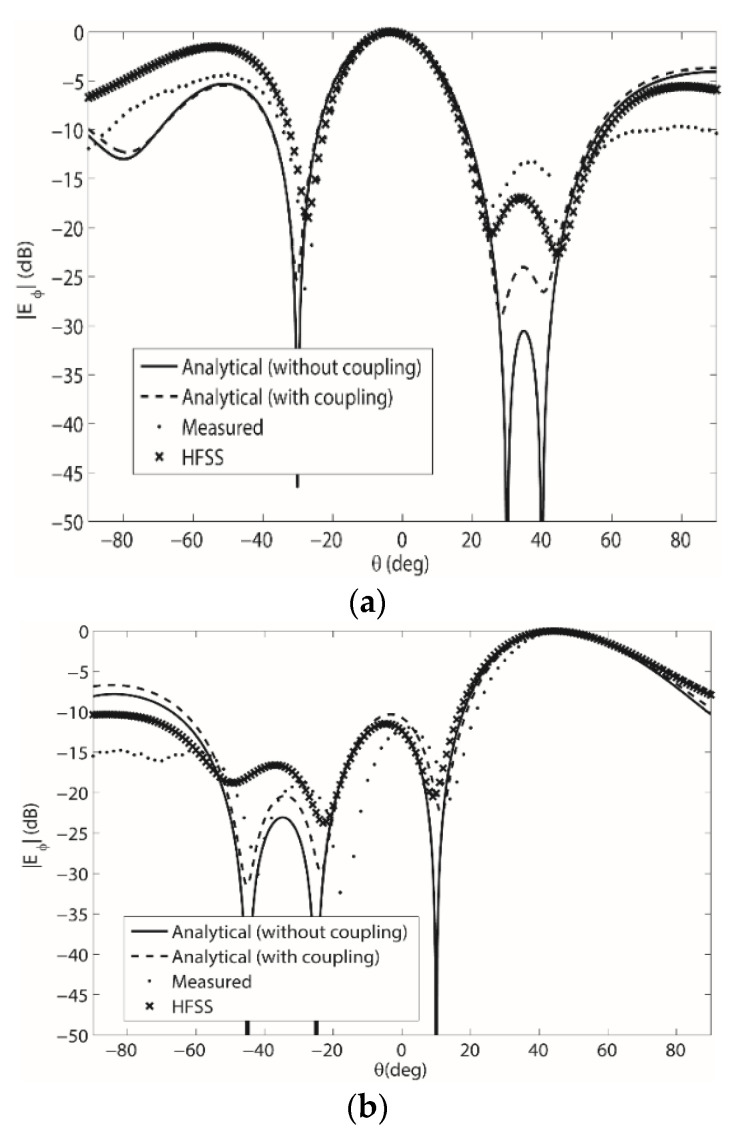
(**a**) Pattern 1 beamforming results for the 1 × 4 microstrip patch array on the singly curved (wedge-shaped) surface with *θ_b_* = 30°; (**b**) Pattern 2 beamforming results for the 1 × 4 microstrip patch array on the singly curved (wedge-shaped) surface with *θ_b_* = 30°.

**Figure 7 sensors-22-06616-f007:**
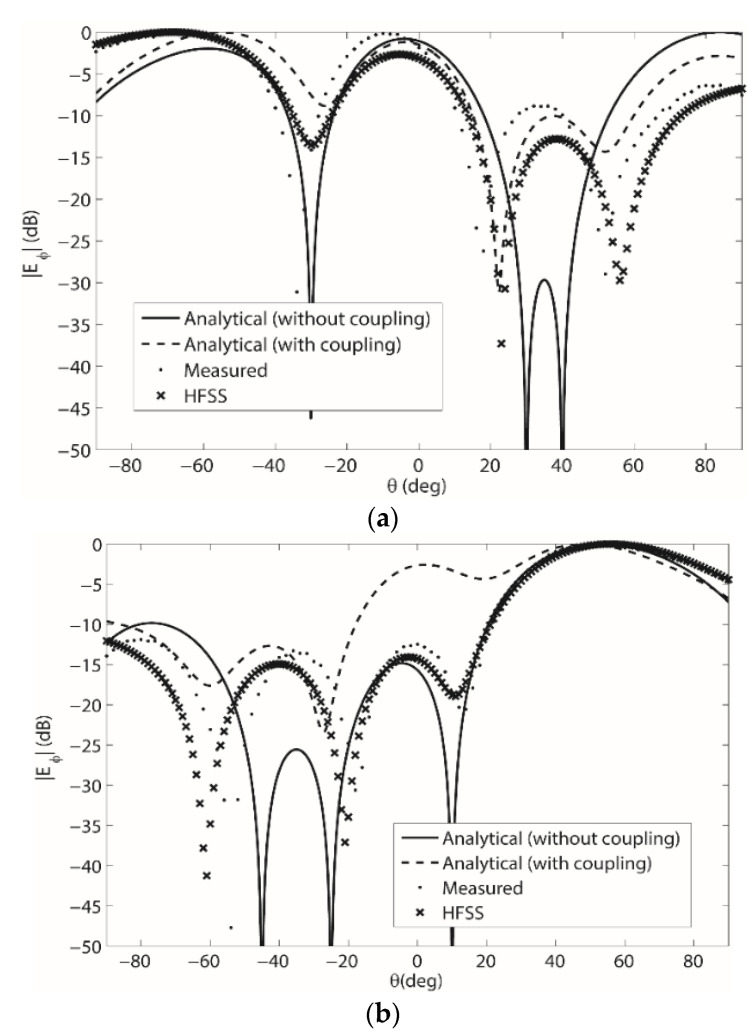
(**a**) Pattern 1 beamforming results for the 1 × 4 microstrip patch array on the singly curved (wedge-shaped) surface with *θ_b_* = 45°; and (**b**) Pattern 2 beamforming results for the 1 × 4 microstrip patch array on the singly curved (wedge-shaped) surface with *θ_b_* = 45°.

**Figure 8 sensors-22-06616-f008:**
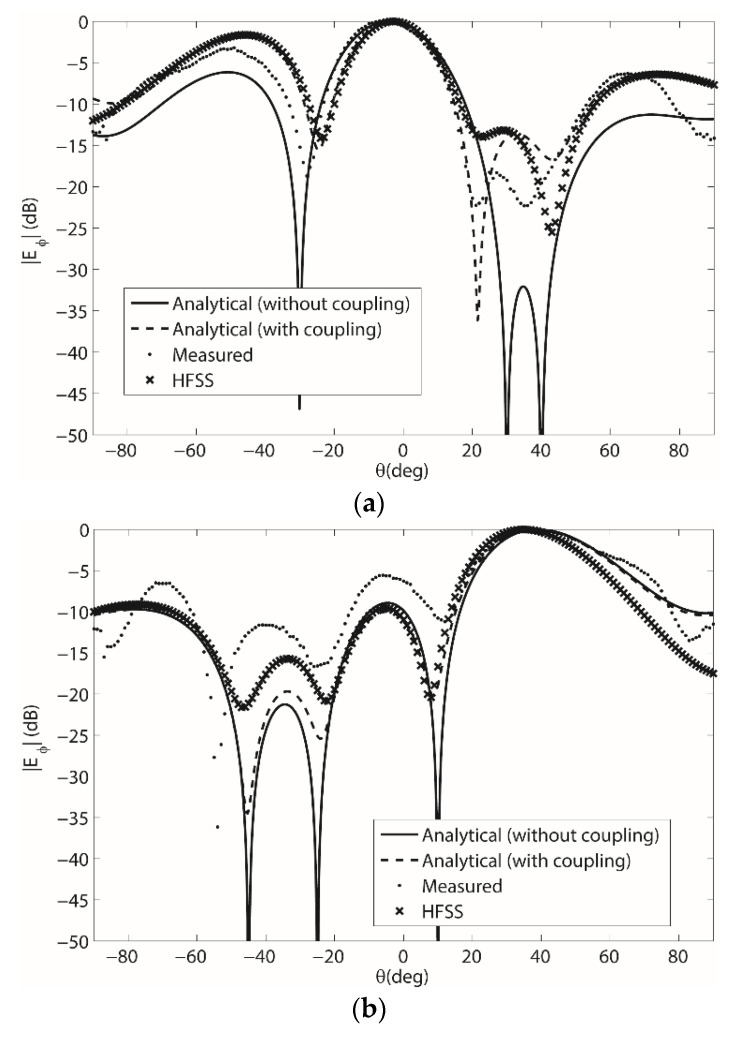
(**a**) Pattern 1 beamforming results for the 1 × 4 microstrip patch array on the cylindrical curvature surface with *r* = 10 cm; (**b**) Pattern 2 beamforming results for the 1 × 4 microstrip patch array on the cylindrical curvature surface with *r* = 10 cm.

**Figure 9 sensors-22-06616-f009:**
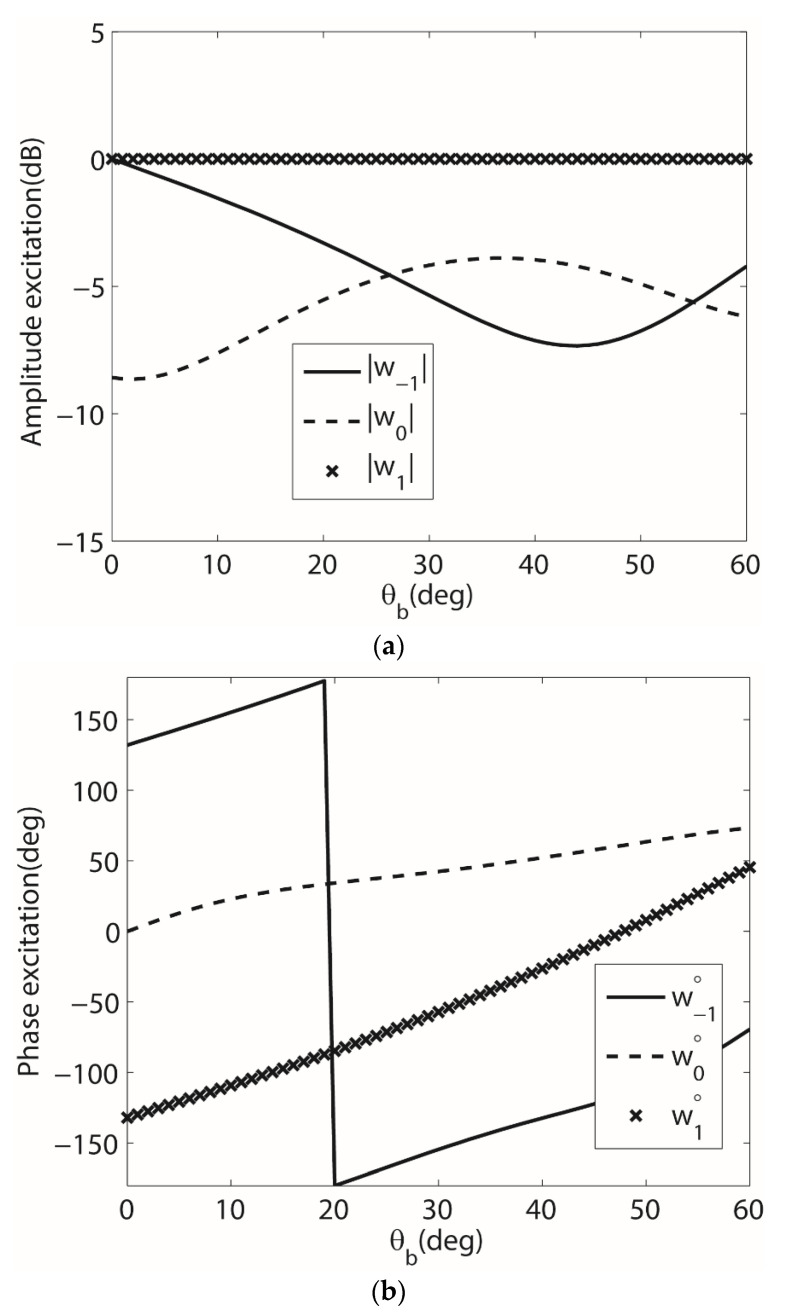
(**a**) Magnitude of the array weights for the 1 × 4 array on the singly curved (wedge) surface and (**b**) Phase of the array weights for the 1 × 4 array on the singly curved (wedge) surface.

**Table 1 sensors-22-06616-t001:** Specifications of the beamforming patterns.

Variable	Pattern 1	Pattern 2
$\theta_{SOI}$	0°	40°
$\theta_{SNOI_{1}}$	−30°	−45°
$\theta_{SNOI_{2}}$	30°	−25°
$\theta_{SNOI_{3}}$	40°	10°

## Data Availability

Not applicable.

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
