# Peer review of "Beamforming with 1 × N Conformal Arrays"

_sensors, 2022, doi:10.3390/s22176616_

Round 1

Reviewer 1 Report

In this paper, the adaptive beamforming patterns of microstrip patch antenna arrays on changing flexible curved surfaces has been developed to compute the desired beamforming amplitudes and phases to excite the antenna elements in the array. Moreover, as a proof of concept, a linear four-element microstrip patch antenna array was embedded on two deformed conformal surfaces to investigate the projection method for desired beamforming patterns. Topic is very interesting. Reviewer has the following minor comments:

  • There are several typos and formatting issues in the paper. For example: page 2, line 62. Authors must revise the paper carefully.
  • Main continuation should be highlighted pointwise.
  • 1, Fig. 2 (a) and Fig. 2 (b) are not clear.
  • Line spacing should be same throughout the paper.

Reviewer 2 Report

In this paper, the adaptive beamforming patterns of microstrip patch antenna arrays on changing flexible curved surfaces are developed to compute the desired beamforming amplitudes and phases to excite the antenna elements in the array. Generally speaking, the work in this paper is not well described. Besides, the simulations lacking of comprehensive discussion are not convincing. Some details about authors’ work should be discussed in detail.

  1. Line 184: when the matrix A is ill-condition, how to get W?
  2. In simulations, the influence of SNR on authors’ method is not comprehensively discussed. At this point, the authors should discuss the influence of SNR on authors’ method.

Reviewer 3 Report

This paper reports interesting results about the beam forming evolution of 1X5 patch antenna array on both wedge and cylinder surfaces. Both simulation and experimental results have been presented with mutual agreement. However there are several concerns:

1. Microstrip patch antenna are widely used in compact systems due to its small size and high efficiency compared with traditional antennas. Here, the design is more like traditional antenna with 3D formation. It is not coplanar anymore. Authors need to clearly highlight the real advantage of this design compared with other previously reported ones. 

2. Also only two special case have been investigated about the 3d configuration of the antenna array. In real application, it is supposed to be on a flexible substrate or not? How to maintain a reasonable gain with a antenna array on a substrate with changing form? Is there any product or POC communication system with this kind of design? 

Round 2

Reviewer 2 Report

The paper has been Improved.

Reviewer 3 Report

Authors have answered my questions. I recommend it for publication.